# Red and Processed Meat Intake, Polygenic Risk Score, and Colorectal Cancer Risk

**DOI:** 10.3390/nu14051077

**Published:** 2022-03-03

**Authors:** Xuechen Chen, Michael Hoffmeister, Hermann Brenner

**Affiliations:** 1Division of Clinical Epidemiology and Aging Research, German Cancer Research Center (DKFZ), 69120 Heidelberg, Germany; xuechen.chen@dkfz-heidelberg.de (X.C.); m.hoffmeister@dkfz-heidelberg.de (M.H.); 2Medical Faculty Heidelberg, Heidelberg University, 69117 Heidelberg, Germany; 3German Cancer Consortium (DKTK), German Cancer Research Center (DKFZ), 69120 Heidelberg, Germany; 4Division of Preventive Oncology, German Cancer Research Center (DKFZ), National Center for Tumor Diseases (NCT), Im Neuenheimer Feld 460, 69120 Heidelberg, Germany

**Keywords:** red and processed meat, colorectal cancer, polygenic risk score, genetic risk equivalent

## Abstract

High red and processed meat intake (RPMI) is an established risk factor for colorectal cancer (CRC). We aimed to assess the impact of RPMI on CRC risk according to and in comparison with genetically determined risk, which was quantified by a polygenic risk score (PRS). RPMI and potential confounders (ascertained by questionnaire) and a PRS (based on 140 CRC-related loci) were obtained from 5109 CRC cases and 4134 controls in a population-based case–control study. Associations of RPMI with CRC risk across PRS levels were assessed using logistic regression models and compared to effect estimates of PRS using “genetic risk equivalent” (GRE), a novel metric for effective risk communication. RPMI multiple times/week, 1 time/day, and >1 time/day was associated with 19% (95% CI 1% to 41%), 41% (18% to 70%), and 73% (30% to 132%) increased CRC risk, respectively, when compared to RPMI ≤ 1 time/week. Associations were independent of PRS levels (p_interaction_ = 0.97). The effect of RPMI > 1 time/day was equivalent to the effect of having 42 percentiles higher PRS level (GRE 42, 95% CI 20–65). RPMI increases CRC risk regardless of PRS levels. Avoiding RPMI can compensate for a substantial proportion of polygenic risk for CRC.

## 1. Introduction

Colorectal cancer (CRC) is a heterogeneous disease resulting from a complex interplay between environmental and genetic risk factors. In 2015, the International Agency for Research on Cancer rated processed meat as carcinogenic to humans (Group 1) and red meat as a probable carcinogen to humans (Group 2A) [1]. It is unclear, however, if and to what extent this relationship with CRC risk may vary with genetic risk of CRC.

So far, genome-wide association studies have identified more than 100 common genetic variants associated with CRC risk [2,3,4,5]. Analyses of the interactions between environmental risk factors and genetic variants are essential for clarifying the mechanisms through which they influence colorectal carcinogenesis and building correct joint models of genetic and environmental risk factors for risk prediction. No significant interactions of red and processed meat with single CRC-related risk variants were observed after adjusting for multiple comparisons in previous studies [6,7,8]. These studies often suffer from very limited power, especially in the light of the harsh penalties of multiple testing and weak main effects of single variants. A polygenic risk score (PRS) built based on multiple CRC-related risk alleles might enable a more powerful assessment of gene–environment interactions. However, whether the effects of red and processed meat intake on CRC risk vary according to PRS levels remains to be determined. Another important issue that needs to be addressed is that risk communication with respect to interplay between environmental and genetic risk factors is so complex that traditional measures like odds ratios (ORs), which are hard to communicate even for risk factor main effects [9,10], might be of very limited use in this context.

In this large population-based study from Germany, we aimed to assess the association of red and processed meat intake with CRC risk across various levels of a PRS that incorporates information from 140 independent genetic variants on CRC risk [4]. Furthermore, we aimed to compare the effects of red and processed meat intake with the impact of genetic predisposition using “genetic risk equivalent” (GRE) [11,12], a recently developed metric aimed at enhancing effective risk communication.

## 2. Materials and Methods

### 2.1. Study Design and Study Population

Data were extracted from the DACHS (Darmkrebs: Chancen der Verhütung durch Screening (German); Colorectal Cancer: Chances for Prevention Through Screening (English)) Study, an ongoing large population-based case–control study of the German Cancer Research Center in Heidelberg, Germany. Details of the DACHS study have been reported elsewhere [13,14]. Briefly, since 2003, patients with a histologically confirmed and first diagnosed primary CRC are eligible if they are 30 years or older (no upper age limit), can speak German, and are physically and mentally able to participate in an approximately one-hour interview. Controls without a history of CRC are randomly selected from population registries by frequency matching with respect to age, sex, and county of residence. This analysis was based on participants recruited from 2003 to 2017 for whom genetic data and environmental risk data were available.

The DACHS study was approved by the ethics committee of the Heidelberg Medical Faculty of Heidelberg University and the state medical boards of Baden-Wuerttemberg and Rhineland-Palatinate. Written informed consent was obtained from each participant.

### 2.2. Data Collection

Information regarding sociodemographic, medical, and lifestyle histories was collected by trained interviewers using a standardized questionnaire. Interviews were usually scheduled in hospital for cases and at home for controls. Additionally, discharge letters and pathology reports were collected for all cases. Buccal swabs and blood samples were taken for both cases and controls. A minority of control participants that opted out of the interview provided some key information in a short self-administered questionnaire and were excluded from this analysis, since detailed dietary and genetic data were not collected from these participants.

### 2.3. Dietary Assessment

Dietary information was obtained using a 23-item food frequency questionnaire at baseline. Participants were asked about their average frequency of consumption over the previous 12 months before the date of the interview (controls) or the date of the diagnosis (cases). There were six possible responses for each item: never; less than once per week; once per week; multiple times per week; once per day; multiple times per day. As outlined in Appendix A, we combined the information on frequency of red and processed meat consumption into a common variable with the following four categories: ≤1 time/week (reference group), multiple times/week, 1 time/day, and >1 time/day. The same categories were used in a previously published study [15] which used the data from 2003–2010 based on the DACHS study to explore the associations of meat intake with major molecular pathological features of CRC. Other main food groups including fish, whole grains (bread, cereals, or others), vegetables/salad, fruits, and dairy foods (cheese/curd, yogurt, or milk), which were considered as covariates in the analyses, were reclassified into two reasonably sized groups for consideration as covariates in the analysis. Details of diet information have also been described in previous studies [15,16,17].

### 2.4. Derivation of Polygenic Risk Score

DNA was extracted from blood samples (in 99.1% of participants) or from buccal cells (in 0.9% of participants). Information about genotyping and imputation of missing genotypes is provided in Appendix A. The PRS was built based on 140 CRC-related risk loci that were identified in a recent genome-wide association study [4] and extracted from our genetic data (Appendix A). The score was calculated as the sum of risk alleles of the respective variants (0, 1, or 2 copies of the risk allele for genotyped single-nucleotide polymorphisms; imputed dosages for imputed single-nucleotide polymorphisms).

### 2.5. Statistical Analysis

We displayed and compared the distributions of key characteristics among cases and controls using the chi-square tests. The association of red and processed meat intake with CRC risk was assessed using logistic regression models adjusted for the matching factors of age and sex. Additional covariates were selected following a forward and backward elimination approach using the “StepAIC” package [18] in R software and involving all covariates shown in Table 1. The model with the lowest Akaike Information Criteria value was selected for the following analysis. After variable selection, the multivariable models included age, sex, school education (<9/9–10/>10 years of schooling), body mass index about 10 years before enrolment (three categories with cut-offs at 25.0 and 30.0 kg/m^2^), smoking (never/former/current smokers), alcohol consumption (above/below the recommended maximum limits of alcohol consumption of 12 g and 24 g ethanol daily for women and men, respectively [19]), history of colonoscopy (yes/no), history of diabetes (yes/no), family history of CRC (yes/no), current use of statins more than once per week (yes/no), regular use of nonsteroidal anti-inflammatory drugs (NSAIDs) including aspirin at least two times per week for at least one year (yes/no), fish intake (<1 time/week/≥1 time/week), whole grains intake (<1 time/day/≥1 time/day), vegetables intake (<1 time/day/≥1 time/day), fruits intake (<1 time/day/≥1 time/day), dairy foods intake (≤1 time/day/>1 time/day), and the PRS (continuous variable).

To investigate the variations of ORs of meat intake across PRS levels, we conducted association analyses by PRS categories in addition to analyses in the entire study population (classification of PRS: very low, ≤10th percentile; low, 11th–25th percentile; medium, 26th–75th percentile; high, 76th–90th percentile; very high, >90th percentile). We tested the interaction of red and processed meat intake with PRS (continuous or categorical variable) by adding pertinent cross-product terms into the models. Furthermore, joint effects of PRS and meat intake were assessed using participants who were at a medium level of PRS and consumed red and processed meat ≤1 time/week as the uniform reference group.

Site- and stage-specific association analyses were conducted for the subsites of the colon including proximal colon (cecum to left flexure) and distal colon and rectum, and for stages I–III and stage IV. We furthermore conducted stratified analyses of the association of red and processed meat intake with CRC risk by age (≤55/>55), sex (male/female), family history of CRC (yes/no), and history of colonoscopy (yes/no), and we tested for potential interaction by these factors by including product terms between red and processed meat consumption and these covariates in the regression models.

Finally, GREs for red and processed meat categories were calculated as ratios of the regression coefficients for red and processed meat intake and PRS percentiles from logistic regression models using an approach in analogy with the previously developed and well-established concept of risk and rate advancement periods [20]. A description of the calculation of GREs and confidence intervals (CIs) for GREs has been reported previously [11] and is summarized in the Appendix A. This approach allowed for a direct comparison of the impact of environmental risk factors with the impact of genetic predisposition as reflected in the PRS. For example, a GRE of 20 can be interpreted as meaning that the estimated risk factor effect is equivalent to an increased risk by having a level of PRS that is 20 percentiles higher.

All analyses were performed using R software version 4.1.1. Statistical tests were two-sided with an alpha level of 0.05.

## 3. Results

### 3.1. Baseline Characteristics of the Study Population

After excluding participants with missing information of red and processed meat intake (15 cases and 14 controls), 5109 patients with CRC and 4134 controls were included in this study (Figure 1). The distribution of baseline characteristics among cases and controls is presented in Table 1. In total, 60.2% of cases and 61.4% of controls were males. The median age for cases and controls was 69 years and 70 years, respectively. Generally, cases were more likely to have a lower level of education, to consume red and processed meat more frequently, to smoke, to consume higher amounts of alcohol, to have a higher level of physical activity, to be overweight or obese, and to report a history of diabetes and family history of CRC. They reported less frequent use of NSAIDs and statins and were less likely to have had a previous colonoscopy examination, and they consumed whole grains, vegetables, fruits, and dairy foods less frequently than controls.

### 3.2. Association of Red and Processed Meat Intake and PRS with CRC Risk

Table 2 shows the association of red and processed meat intake with CRC risk in the whole study population. Red and processed meat intake multiple times/week, 1 time/day, and >1 time/day was significantly associated with 19%, 41% and 73% increased risk of CRC when compared to the lowest category of red and processed meat intake (≤1 time/week) in the multivariable model. A similarly strong positive association between the frequency of red and processed meat consumption was seen within each category of PRS, even though it did not reach statistical significance in the subgroups with very low and low PRS, given the small numbers of cases in these subgroups (Table 3). Tests for interaction between red and processed meat and PRS did not reach statistical significance (*p* value = 0.97 and 0.79 in models with PRS included as a continuous or categorical variable, respectively).

Results of the joint association of red and processed meat intake and the PRS with CRC risk are presented in Table 4. Compared to the reference group who consumed red and processed meat ≤1 time/week and were at a medium level of PRS, ORs (95% CIs) for CRC risk ranged from 0.30 (0.15–0.58) in the group with the same low consumption but a very low PRS to 3.23 (1.52–7.41) in the group with the highest consumption and a very high PRS.

### 3.3. Genetic Risk Equivalents for Red and Processed Meat Intake Categories

Estimates of GRE for different red and processed meat intake categories are presented in Table 5. Participants who consumed red and processed meat >1 time/day, 1 time/day, or multiple times/week had a GRE (95% CI) of 42 (20–65), 26 (12–40), or 13 (1–26) compared to those who occasionally or never consumed red and processed meat (≤1 time/week). This indicates that these levels of consumption were associated with an increase in CRC risk that was equivalent to having a 42, 26, or 13 percentile higher PRS. Increases in risk and GRE with increasing consumption of red and processed meat were consistently seen for cancers in the proximal and distal colon and the rectum, and for stages I–III and stage IV CRC.

Results for the stratified analyses are summarized in Figure 2 and Appendix A. GREs for those who consumed red and processed meat >1 time/day were particularly high among males (GRE 50, 95% CI 20–80), among participants without a family history of CRC (GRE 52, 95% CI 26–79), and among participants without a history of colonoscopy (GRE 51, 95% CI 22–80), whereas associations were weaker and partly not statistically significant among younger and female participants and those with a family history of CRC or with a previous colonoscopy. However, tests for interaction did not reach statistical significance for any of the subgroup comparisons.

## 4. Discussion

In this large population-based study case–control study, we found that red and processed meat intake was strongly and independently associated with CRC risk, resulting in a very strong risk gradient between those at very high PRS and high consumption of red and processed meat, and those with a very low PRS and low consumption. The effect estimate of red and processed meat consumption in the highest category versus the lowest category was equivalent to that of having a level of PRS 42 percentiles higher, suggesting that abstaining from consuming high levels of red and processed meat might compensate for a large share of genetically increased risk.

Our results are consistent with the accumulated evidence that has linked higher intake of red and processed meat to increased CRC risk [1,15,21,22,23]. The ORs (95% CI) for the highest category vs. lowest category of red and processed meat intake were 1.21 (95% CI 1.09–1.34) for colon cancer and 1.26 (95% CI 1.09–1.45) for rectal cancer in a recent meta-analysis of 38 studies for colorectal, colon, and rectum cancer [23]. Dose–response meta-analysis showed a 19% (95% CI 10% to 30%; 10 studies with 10,010 cases) increased risk of colon cancer and a 17% (95%CI −1% to 39%; 6 studies with 3455 cases) increased risk of rectal cancer for each 100 g/day increase of red and processed meat intake [22]. We likewise did not observe variations of associations of red and processed meat intake with CRC risk by cancer site. Subgroup analyses by age, sex, and family history pointed to potential differences in the strength of the association between red and processed meat consumption, but statistical tests for interaction did not reach statistical significance. Even larger studies are needed to evaluate potential interactions of these factors with adequate power.

Several substances in red and processed meat have been suggested to have carcinogenic effects, such as preservatives (e.g., inorganic sulfur, nitrates, and nitrites), some nutrients enriched in meats (e.g., heme iron, sulfur-containing amino acids, and saturated fats), and certain chemicals present during meat processing and cooking (e.g., heterocyclic amines and polycyclic aromatic hydrocarbons) [24,25,26,27]. For example, a recent in vitro study showed that heme iron present in red meat might induce CRC risk through DNA adduct formation [24]. Additionally, gut microbiota might be one of the key factors that mediate or modify the effects of red and processed meat on CRC risk [28,29].

Gene–environment interaction studies might help to better understand and more fully disclose the role of red meat and processed meat intake in CRC risk. No significant interactions with single CRC-related genetic risk variants were found after correction for multiple testing in previous studies [6,7], possibly reflecting a lack of power to detect such interactions even in very large-scale international consortia. Analyses of interactions by PRS, reflecting cumulative genetic burden based on a set of disease-related loci and showing a robust monotonic association with CRC risk, could help to resolve the issues of weak main effects of single loci and multiple testing penalties in gene–environment interaction studies [30]. To our knowledge, only one previous study, conducted by Yang et al., assessed interactions of red meat (times/week) and processed meat intake (times/week) with the PRS (continuous variable). In this study from the UK, red and processed meat were analyzed separately, and no statistically significant interactions were found [8]. We likewise did not observe significant interactions despite a larger number of cases and a combined variable based on information from red meat and processed meat intake, which showed a stronger association with CRC risk. We extend their results by exploring the associations of meat intake with CRC risk across various levels of genetically determined CRC risk in the whole population and in detailed subgroup analyses, and comparing the risk from meat intake to the risk from predetermined polygenic risk for CRC.

Both the study by Yang et al. and our study evaluated interactions on the multiplicative scale that can be directly estimated from logistic regression models. Although no such interactions were detected, it should be noted that the same relative risk of meat intake across PRS levels implies a higher increase in absolute risk of most frequent meat intake among people with higher PRS levels, due to their higher “baseline risk” [17]. Thus, people with high PRS levels might potentially benefit most from avoiding red and processed meat intake. Our findings underscore the importance of reduced red and processed meat intake, in particular for those with high susceptibility to CRC, and also provide insights related to joint consideration of the PRS and meat intake for CRC risk stratification.

To our knowledge, no previous study has directly compared the estimated impact of high red and processed meat consumption with predetermined polygenic risk to CRC. We compared the effect estimate of red and processed meat intake to the effect estimate of PRS using the novel concept of genetic risk equivalents, which expresses effect estimates for environmental or lifestyle risk factors in terms of percentile differences of genetic risk. Demonstrating that avoidance of such risk factors could potentially compensate for a substantial share of increased genetic risk might be helpful in communicating risk estimates and the potential for prevention. The large GRE for the group with the most frequent intake of red and processed meat estimated in our study underlines the importance of this dietary factor for CRC risk and underlines and illustrates the large preventive potential of lower consumption of red and processed meat.

Our study has several strengths. The DACHS study is one of the largest population-based case–control studies on CRC. The comprehensive information provided by the study participants along with comprehensive genotyping enables thorough adjustment for potential confounders and interaction testing in multivariable analyses. In particular, it strongly extends the still very limited evidence on the potential interaction between red and processed meat intake and the PRS with respect to CRC risk. Furthermore, to the best of our knowledge, this is the first study communicating the risk of red and processed meat intake in terms of the equivalent difference in background genetic risk.

There are also some limitations that merit consideration. First, ascertainment of red and processed meat intake was based on self-reports by a food frequency questionnaire with a limited number of items in each food group. Furthermore, only frequencies but not the amount of consumption were ascertained, which, along with imperfect reporting, is expected to have resulted in some misclassification and precluded more precise analyses of dose–response relationships. Second, controls less frequently consumed red and processed meat and thus might be more health conscious compared to cases with CRC. Additionally, only interviewed control participants who provided detailed information were included in the analyses. These points imply the potential of selection bias, which could have led to some overestimation of ORs and the corresponding GREs for red and processed meat intake, although we adjusted for some proxy measures of health consciousness such as education, history of colonoscopy, and fruit intake to reduce the potential influence of such bias in the current analysis. Third, although we controlled for a range of potential confounders, residual confounding related to CRC risk and meat intake may still be present. Fourth, this study was based on a Caucasian population and thus results need to be validated in other ethnic populations with other ethnicities and different genetic backgrounds, natural environments, and food consumption cultures. Fifth, we were unable to conduct analyses for specific CRC subtypes, given that tumor subtype information was only available for a subgroup of the study population and study power was thereby limited to address potential interactions of PRS and red and processed meat intake on the risk of specific subtypes of CRC [15]. Further studies are needed to follow up and address this important issue.

## 5. Conclusions

In conclusion, our study provides evidence of a major contribution of frequent red and processed meat intake to increased CRC risk. In relative terms, this risk increase seems to be similar across various levels of PRS. At the same time the absence of multiplicative interaction implies that in absolute terms, the excess risk associated with frequent red and processed meat consumption is expected to be highest among those with higher genetic risk, who might therefore benefit most from restricting their red and processed meat consumption. The large GREs estimated for frequent red and processed meat consumption suggest that such restriction could compensate for a large share of increased genetic risk, which might be helpful in risk communication and preventive efforts. Further research should validate our results in populations with different ethnic backgrounds and different cultures and lifestyle habits.

## Figures and Tables

**Figure 1 nutrients-14-01077-f001:**
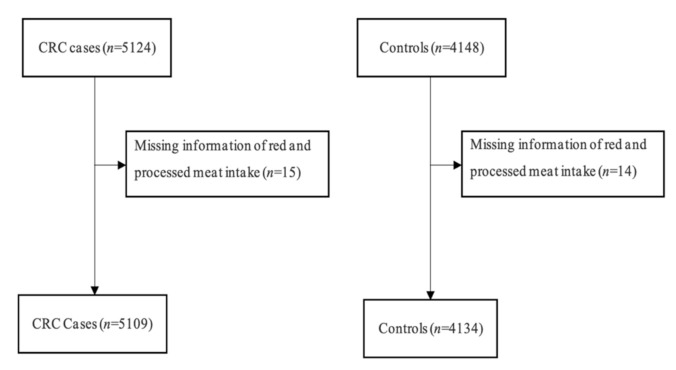
Flowchart of inclusion of study participants. Abbreviation: CRC, colorectal cancer.

**Figure 2 nutrients-14-01077-f002:**
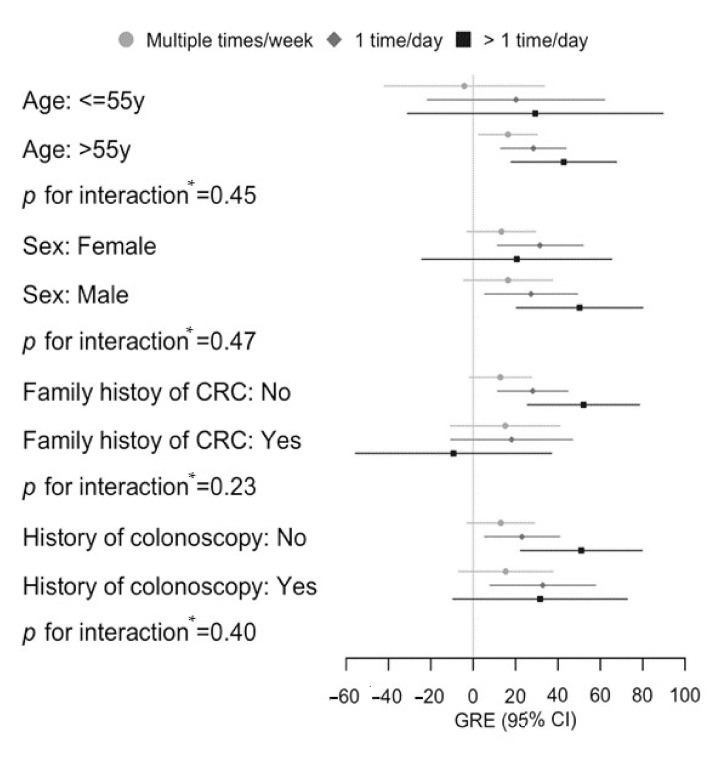
Genetic risk equivalents for comparisons between red and processed meat intake categories in different subgroups. * Interactions were tested by additionally including the multiplicative term of stratification factors (age, sex, family history of CRC, or history of colonoscopy) and red and processed meat intake in the multivariable models, but with a polygenic risk score included as percentiles (per 10 percentiles, continuous variable). Participants who consumed red and processed meat ≤1 time/week were used as reference in each subgroup. Abbreviations: CI, confidence interval; CRC, colorectal cancer; GRE, genetic risk equivalent.

**Table 1 nutrients-14-01077-t001:** Baseline characteristics of colorectal cancer patients and controls.

Characteristics	Cases, *N* (%)	Controls, *N* (%)	*p* Value ^6^
Total	5109	4143	
Sex: Male	3077 (60.2)	2540 (61.4)	
Age: Median (Q1, Q3)	69 (62, 76)	70 (62, 76)	
School education (years)			
<9	3336 (65.3)	2280 (55.2)	
9–10	906 (17.7)	877 (21.2)	<0.0001
>10	858 (16.8)	970 (23.5)	
Red and processed meat intake ^1^			
≤1 time/week	404 (7.9)	482 (11.7)	
multiple times/week	3067 (60.0)	2514 (60.8)	
1 time/day	1411 (27.6)	1010 (24.4)	<0.0001
>1 time/day	227 (4.4)	128 (3.1)	
Smoking status			
Never	2283 (44.7)	2088 (50.5)	
Former	2039 (39.9)	1586 (38.4)	<0.0001
Current	767 (15.0)	449 (10.9)	
Alcohol consumption ^2^			
Above recommended threshold	1318 (25.8)	936 (22.6)	<0.001
Physical activity (MET-hour/week) ^3^			
Q1 (≤121.6)	1145 (22.4)	1032 (25.0)	
Q2 (121.7–178.5)	1247 (24.4)	1030 (24.9)	
Q3 (178.6–244.7)	1234 (24.2)	1030 (24.9)	0.0029
Q4 (>244.7)	1421 (27.8)	1031 (24.9)	
BMI (kg/m^2^, about 10 years before enrolment)			
<25	1537 (30.1)	1578 (38.2)	
25–<30	2365 (46.3)	1875 (45.4)	<0.0001
30+	1137 (22.3)	649 (15.7)	
History of diabetes	971 (19.0)	559 (13.5)	<0.0001
Family history of colorectal cancer	738 (14.4)	450 (10.9)	<0.0001
Use of NSAIDs ^4^	1453 (28.4)	1572 (38.0)	<0.0001
Use of statins ^5^	874 (17.1)	927 (22.4)	<0.0001
History of colonoscopy	1359 (26.6)	2493 (60.3)	<0.0001
Fish < 1 time/week ^1^	1562 (30.6)	1191 (28.8)	0.065
Whole grains < 1 time/day ^1^	3257 (63.8)	2202 (53.3)	<0.0001
Vegetables < 1 time/day ^1^	4243 (83.0)	3108 (75.2)	<0.0001
Fruits < 1 time/day ^1^	1907 (37.3)	1223 (29.6)	<0.0001
Dairy foods ≤ 1 time/day ^1^	2906 (56.9)	2059 (49.8)	<0.0001

NOTE: Missing values for cases/controls: school education 9/7, smoking status 20/11, alcohol consumption 15/14, physical activity 62/11, BMI 70/32, history of diabetes 7/5, family history of colorectal cancer 3/3, use of statins 2/5, fish 8/3, whole grains 6/6, vegetables 9/3, fruits 13/7, dairy foods 27/20. ^1^ Average frequency of consumption in the previous 12 months before interview (controls) or diagnosis (cases). ^2^ Lifetime average daily alcohol consumption, measured in grams of ethanol; the recommended daily limits are 12 g and 24 g ethanol for women and men, respectively. ^3^ Lifetime average physical activity, measured in MET-hour/week and categorized according to the distribution of physical activity among controls. ^4^ Use of NSAIDs is defined as taking NSAIDs (including aspirin) at least 2 times a week for at least 1 year. ^5^ Use of statins is defined as current use of statins more than once a week. ^6^ Cases and controls were matched for age and sex during the recruitment, so *p* values were not reported for age and sex. Abbreviations: BMI, body mass index; MET, metabolic equivalent of task; NSAID, nonsteroidal anti-inflammatory drug; Q, quartile.

**Table 2 nutrients-14-01077-t002:** Association of red and processed meat intake with colorectal cancer risk in the whole study population.

Red and Processed Meat Intake	Cases, N (%)	Controls, N (%)	OR (95% CI) ^1^	OR (95% CI) ^2^
≤1 time/week	387 (7.9)	469 (11.7)	Ref.	Ref.
Multiple times/week	2925 (59.8)	2444 (60.9)	1.49 (1.29, 1.73)	1.19 (1.01, 1.41)
1 time/day	1358 (27.8)	979 (24.4)	1.76 (1.49, 2.06	1.41 (1.18, 1.70)
>1 time/day	222 (4.5)	124 (3.1)	2.27 (1.75, 2.95)	1.73 (1.30, 2.32)
*p* value for interaction ^3^				0.97

^1^ Adjusted for age and sex. ^2^ Additionally adjusted for school education, body mass index, smoking, alcohol consumption, history of colonoscopy, history of diabetes, family history of colorectal cancer, use of statins, use of nonsteroidal anti-inflammatory drugs, fish, whole grains, vegetables, fruits, dairy foods, and polygenic risk score (continuous variable). ^3^ Interactions were tested by including a cross-product term of red and processed meat intake and polygenic risk score (continuous variable) in multivariable models. Abbreviations: CI, confidence interval; OR, odds ratio; Ref., reference.

**Table 3 nutrients-14-01077-t003:** Association of red and processed meat intake with colorectal cancer risk by polygenic risk score.

PRS ^1^		Red and Processed Meat Intake				
		≤1 Time/Week	Multiple Times/Week	1 Time/Day	>1 Time/Day	Per Category Increase
Very low	Cases, N (%)	13 (6.0)	128 (59.3)	65 (30.1)	10 (4.6)	
	Controls, N (%)	48 (11.9)	232 (57.7)	105 (26.1)	17 (4.2)	
	OR (95% CI) ^2^	Ref.	2.07 (1.11, 4.12)	2.32 (1.18, 4.80)	2.35 (0.85, 6.49)	1.27 (0.99, 1.62)
	OR (95% CI) ^3^	Ref.	1.64 (0.81, 3.51)	1.83 (0.86, 4.08)	1.82 (0.57, 5.80)	1.19 (0.90, 1.57)
Low	Cases, N (%)	41 (8.3)	290 (58.7)	136 (27.5)	27 (5.5)	
	Controls, N (%)	70 (11.6)	376 (62.0)	143 (23.6)	17 (2.8)	
	OR (95% CI) ^2^	Ref.	1.34 (0.88, 2.05)	1.66 (1.05, 2.65)	2.78 (1.36, 5.84)	1.33 (1.11, 1.59)
	OR (95% CI) ^3^	Ref.	0.96 (0.59, 1.56)	1.12 (0.66, 1.91)	1.70 (0.76, 3.87)	1.16 (0.94, 1.43)
Medium	Cases, N (%)	209 (8.7)	1433 (59.5)	663 (27.5)	104 (4.3)	
	Controls, N (%)	228 (11.3)	1229 (61.2)	488 (24.3)	64 (3.2)	
	OR (95% CI) ^2^	Ref.	1.30 (1.06, 1.59)	1.53 (1.22, 1.92)	1.83 (1.27, 2.65)	1.21 (1.11, 1.33)
	OR (95% CI) ^3^	Ref.	1.05 (0.83, 1.31)	1.26 (0.98, 1.62)	1.39 (0.93, 2.08)	1.15 (1.04, 1.27)
High	Cases, N (%)	61 (6.6)	555 (60.3)	259 (28.2)	45 (4.9)	
	Controls, N (%)	73 (12.2)	365 (60.9)	145 (24.2)	16 (2.7)	
	OR (95% CI) ^2^	Ref.	1.92 (1.32, 2.78)	2.29 (1.53, 3.46)	3.57 (1.85, 7.18)	1.40 (1.19, 1.65)
	OR (95% CI) ^3^	Ref.	1.69 (1.12, 2.56)	2.06 (1.31, 3.24)	2.63 (1.27, 5.65)	1.34 (1.12, 1.61)
Very high	Cases, N (%)	63 (7.4)	519 (60.8)	235 (27.5)	36 (4.2)	
	Controls, N (%)	50 (12.5)	242 (60.5)	98 (24.5)	10 (2.5)	
	OR (95% CI) ^2^	Ref.	1.85 (1.22, 2.78)	2.14 (1.36, 3.38)	3.33 (1.53, 7.80)	1.37 (1.14, 1.66)
	OR (95% CI) ^3^	Ref.	1.51 (0.94, 2.43)	1.65 (0.98, 2.77)	2.80 (1.16, 7.18)	1.26 (1.02, 1.56)
*p* value for interaction = 0.79 ^4^						

^1^ Classification of PRS: very low, ≤10th percentile; low, 11th–25th percentile; medium, 26th–75th percentile; high, 76th–90th percentile; very high, >90th percentile. ^2^ Adjusted for age and sex; ^3^ Additionally adjusted for school education, body mass index, smoking, alcohol consumption, history of colonoscopy, history of diabetes, family history of colorectal cancer, use of statins, use of nonsteroidal anti-inflammatory drugs, fish, whole grains, vegetables, fruits, dairy foods. ^4^ Interactions were tested by including a cross-product term of red and processed meat intake and PRS (categorical variable) in multivariable models. Abbreviations: CI, confidence interval; OR, odds ratio; PRS, polygenic risk score; Ref., reference.

**Table 4 nutrients-14-01077-t004:** Joint association of red and processed meat intake and polygenic risk score with colorectal cancer risk.

PRS ^1^	Red and Processed Meat Intake			
	≤1 Time/Week	Multiple Times/Week	1 Time/Day	>1 Time/Day
	OR (95% CI) ^2^	OR (95% CI) ^2^	OR (95% CI) ^2^	OR (95% CI) ^2^
Very low	0.30 (0.15, 0.58)	0.49 (0.36, 0.67)	0.58 (0.39, 0.87)	0.50 (0.20, 1.20)
Low	0.66 (0.41, 1.05)	0.68 (0.52, 0.89)	0.79 (0.56, 1.10)	1.20 (0.61, 2.42)
Medium	Ref.	1.05 (0.84, 1.32)	1.28 (1.00, 1.64)	1.42 (0.95, 2.12)
High	0.83 (0.54, 1.27)	1.38 (1.07, 1.78)	1.71 (1.26, 2.32)	2.34 (1.24, 4.61)
Very high	1.49 (0.95, 2.36)	1.96 (1.50, 2.56)	2.08 (1.49, 2.90)	3.23 (1.52, 7.41)

^1^ Classification of PRS: very low, ≤10th percentile; low, 11th–25th percentile; medium, 26th–75th percentile; high, 76th–90th percentile; very high, >90th percentile. ^2^ Models were adjusted for age, sex, school education, body mass index, smoking, alcohol consumption, history of colonoscopy, history of diabetes, family history of CRC, use of statins, use of nonsteroidal anti-inflammatory drugs, fish, whole grains, vegetables, fruits, dairy foods. Abbreviations: CI, confidence interval; OR, odds ratio; PRS, polygenic risk score; Ref., reference.

**Table 5 nutrients-14-01077-t005:** Genetic risk equivalents for comparisons between red and processed meat intake categories for colorectal cancer risk.

	Red and Processed Meat Intake			
	≤1 Time/Week	Multiple Times/Week	1 Time/Day	>1 Time/Day
Controls, N (%)	469 (11.7)	2444 (60.9)	979 (24.4)	124 (3.1)
Cases (All), N (%)	387 (7.9)	2925 (59.8)	1358 (27.8)	222 (4.5)
OR (95% CI) ^1^	Ref.	1.19 (1.01, 1.40)	1.41 (1.18, 1.69)	1.74 (1.31, 2.33)
GRE (95% CI)	Ref.	13.3 (0.6, 26.0)	26.2 (12.0, 40.4)	42.3 (19.6, 64.9)
Cases (Colon) ^2^, N (%)	256 (8.6)	1812 (60.8)	782 (26.3)	128 (4.3)
OR (95% CI) ^1^	Ref.	1.17 (0.98, 1.41)	1.32 (1.08, 1.61)	1.76 (1.28, 2.43)
GRE (95% CI)	Ref.	12.8 (−2.3, 28.0)	22.7 (5.9, 39.6)	46.3 (19.1, 73.4)
Cases (Proximal colon), N (%)	146 (8.8)	1027 (62.0)	420 (25.4)	63 (3.8)
OR (95% CI) ^1^	Ref.	1.28 (1.03, 1.60)	1.40 (1.10, 1.78)	1.73 (1.17, 2.56)
GRE (95% CI)	Ref.	21.8 (2.1, 41.5)	29.7 (7.8, 51.6)	48.4 (12.9, 83.9)
Cases (Distal colon), N (%)	110 (8.3)	782 (59.3)	362 (27.4)	65 (4.9)
OR (95% CI) ^1^	Ref.	1.06 (0.83, 1.37)	1.22 (0.93, 1.61)	1.76 (1.17, 2.65)
GRE (95% CI)	Ref.	4.2 (−13.6, 21.9)	14.2 (−5.3, 33.7)	40.4 (10.5, 70.4)
Cases (Rectum), N (%)	131 (6.8)	1113 (58.2)	576 (30.1)	94 (4.9)
OR (95% CI) ^1^	Ref.	1.26 (1.00, 1.60)	1.64 (1.28, 2.12)	1.82 (1.23, 2.67)
GRE (95% CI)	Ref.	17.6 (−0.6, 35.9)	37.8 (17.4, 58.1)	45.7 (15.5, 75.9)
Cases (Stages I–III), N (%)	334 (8.0)	2488 (59.9)	1144 (27.5)	187 (4.5)
OR (95% CI) ^1^	Ref.	1.18 (0.99, 1.40)	1.37 (1.14, 1.66)	1.76 (1.31, 2.38)
GRE (95% CI)	Ref.	12.6 (−0.6, 25.8)	24.0 (9.4, 38.7)	43.1 (19.7, 66.6)
Cases (Stage IV), N (%)	49 (7.1)	410 (59.0)	203 (29.2)	33 (4.7)
OR (95% CI) ^1^	Ref.	1.31 (0.94, 1.85)	1.66 (1.16, 2.40)	1.80 (1.04, 3.07)
GRE (95% CI)	Ref.	20.6 (−5.6, 46.8)	38.7 (9.7, 67.7)	44.9 (2.5, 87.2)

^1^ Adjusted for age, sex, school education, body mass index, smoking, alcohol consumption, history of colonoscopy, history of diabetes, family history of colorectal cancer, use of statins, use of nonsteroidal anti-inflammatory drugs, fish, whole grains, vegetables, fruits, dairy foods, and polygenic risk score (per 10 percentiles, continuous variable). ^2^ Missing information on anatomic sites (proximal or distal colon) for three colon cancer cases. Abbreviations: CI, confidence interval; GRE, genetic risk equivalent; OR, odds ratio; Ref., reference.

## Data Availability

Data described in the manuscript, code book, and analytic code will be made available upon request from the corresponding author.

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
