# Peer review of "Red and Processed Meat Intake, Polygenic Risk Score, and Colorectal Cancer Risk"

_nutrients, 2022, doi:10.3390/nu14051077_

Round 1

Reviewer 1 Report

This is an interesting manuscript on a topic of wide interest. There have been many publications indicating a relationship between the consumption of red and processed meat and the incidence of colorectal cancer. The authors have studied the individual and combined effects of genetic and dietary contributions to colorectal cancer risk. The study appears to have been carefully conducted and is well described. At the end of their discussion the authors note the limitations of the study including the reliance on self reported food intake frequency rather than a record of measured food consumption. However, this study appears to be “the first study communicating the risk of red and processed meat intake in terms of the equivalent difference in background genetic risk”. That certainly justifies publication.

There are two minor suggestions. It would be helpful to the reader if Figure 2 and its legend could be on the same page. As presented it looks as though the data in Figure 2 are an extension of Table 5. References 8 and 29 are the same reference. Reference 29 could be deleted and could be presented as Reference 8 in the text.

Author Response

Point 1: It would be helpful to the reader if Figure 2 and its legend could be on the same page. As presented it looks as though the data in Figure 2 are an extension of Table 5.

  • Response 1: Thanks for your advice. We have moved Figure 2 as you suggested, and now it is on the same page with its legend. We also formated Table 1 and Table 3 to make sure they fit on one page.

Point 2: References 8 and 29 are the same reference. Reference 29 could be deleted and could be presented as Reference 8 in the text.

  • Response 2: Thanks for alerting us of this oversight. Reference 29 has been deleted.

Reviewer 2 Report

This manuscript by Xuechen Chen et al. describes findings of their assessment regarding the impact of high red and processed meat intake (RPMI) on the risk of colorectal cancer (CRC) according to or in comparison with genetically determined risk that quantified by a polygenic risk score (PRS). Their conclusion includes that RPMI increases CRC risk regardless of PRS levels. The authors also speculate that avoiding RPMI can compensate for a substantial proportion of polygenic risk for CRC.
The findings are of interest. I have only a few comments listed below.
1)    If the authors possess data on association between RPMI/PRS and histopathology of CRC. Please provide them and discuss.
2)    Again, if the authors possess data on association between RPMI/PRS and site (right or left colon) of CRC. Please provide them and discuss.
3)    In this regards, two papers (PMID: 9242481 and PMID: 29458163) can be cited and discussed:

Author Response

Point 1: If the authors possess data on association between RPMI/PRS and histopathology of CRC. Please provide them and discuss.

  • Response 1: We agree that this would be of great interest. Unfortunately, tumor subtype information was only available for a subgroup of the study population which hindered conducting analyses of potential interactions of RPMI and PRS, our main study question, for specific CRC subtypes with reasonable power. We address this important point in the limitations section (Page 11, Lines 364-368).

Point 2: Again, if the authors possess data on association between RPMI/PRS and site (right or left colon) of CRC. Please provide them and discuss.

  • Response 2: Thanks for your suggestion. We now provide such data in Table 5 and refer to them in the text (Page 3, line 132; Page 7; Lines 230-231).

Point 3: In this regards, two papers (PMID: 9242481 and PMID: 29458163) can be cited and discussed:

  • Response 3: Thanks for your advice. These two papers have been added as reference 21 (Page 9, Line 272) and as reference 24 (on page 10, Lines 291-293), respectively.